# Effect of the APD Receiver's Tilted Angle on Channel Capacity for Underwater Wireless Optical Communications

**Meiyan Ju, Hongqiang Shi, Yueheng Li \*, Ping Huang and Guoping Tan**

College of Computer and Information, Hohai University, Nanjing 211100, China; jmy@hhu.edu.cn (M.J.); imshq@hhu.edu.cn (H.S.); huangpinghope@hhu.edu.cn (P.H.); gptan@hhu.edu.cn (G.T.)
\* Correspondence: yueheng_li@hhu.edu.cn

**Abstract:** This paper focuses on investigating the effect of the receiver's tilted angle on the channel capacity of an underwater wireless optical communication (UWOC) system, in which an avalanche photodiode (APD) detector is adopted as the receiver. Under the non-negativity, peak power, and average power constraints, the lower bounds on the capacity of UWOC are derived in detail according to different average-to-peak power ratios. With modeling achieving the maximum of the lower bounds of the capacity as an optimization object, we prove that the proposed optimization issue is in fact a simple convex optimization about the tilted angle of the APD receiver, and then present related theoretical solution for it. Both theoretical analysis and simulation results show that by appropriately tilting the receiver, we can significantly enhance the final capacity performance of the UWOC with APD receiver.

**Keywords:** UWOC; APD receiver; pointing error; capacity bound; convex optimization





## 1. Introduction

Underwater wireless optical communication (UWOC) has attracted great attention in recent years due to its high data rate, low latency, and solid reliability compared with other traditional underwater communication ways, for example, the acoustic and radio frequency (RF) communications [1,2]. Unfortunately, in UWOC, the received signal suffers severe attenuation caused by the optical properties of underwater channel, namely, absorption and scattering, which is defined as channel loss in [3,4] and inevitably degrades the system performance. Besides this, the optical beam needs also to be highly directive for a successful communication. Practically, however, due to the misalignment between the transmitter and receiver, the so-called pointing loss is incurred [4,5]. Obviously, the pointing loss effect will impair the system performance.

So far, many studies have been conducted on the effect of misalignment of the transmitter and receiver on the received intensity [4–7]. However, in the above literature, the receiver planes were fixed and could not be inclined, which greatly limited the performance of the UWOC especially in a dynamic ocean environment. Although [8,9] allowed for the tilting or movement of the receiver, no further considerations were given to theoretically optimize the performance of UWOC by tilting the receiver to overcome the deleterious pointing loss.

Channel capacity is an important indicator for evaluating the performance of communication links. A closed-form expression for the average channel capacity of UWOC was studied based on the anisotropic ocean turbulence channel with Málaga fading in [10]; Reference [11] investigated both the capacity and bit-error-rate (BER) of underwater wireless optical links under weak and strong turbulence by deriving the expressions of the average capacity and BER. In [12], based on modeling the statistical characteristic of the pointing error's variance, the average channel capacity over weak turbulence distribution was established. However, all the literature mentioned above only discussed the channel capacity of UWOC in a traditionally input-independent noise scenario, that is, they adopted

a positive-intrinsic-negative (PIN) photodiode as the receiver and did not consider the effect of other more powerful photodiode, for instance, an avalanche photodiode (APD), on the channel capacity of UWOC in practical application. Since APD can greatly outperform the traditional PIN diode due to its intrinsic average gain [13], people prefer to use APD instead of PIN to mitigate the channel loss and then to enhance the system transmission length in seawater [14]. Despite this, however, the signal model of APD is a little difficult to be analyzed because an excessive shot noise term relating to the input signal is introduced in signal modeling, and this complicates the performance analysis. Although References [15,16] investigated the performance bounds of the channel capacity when the input-dependent noise term is introduced in a free space optical (FSO) communication situation, the key channel loss factor in the transmitting signal construction was neglected to simplify the system modeling and performance analysis. Obviously, this makes the receiving signal model of APD receiver in [15,16] incomplete and the results about the capacity bounds derived in them could not be used directly.

In fact, there are other effective ways to improve the performance of an optical communication system reported in recent literature. For example, Reference [17] revealed that people can use multiple apertures technique at both the transmitter and receiver sides to greatly enhance the transmission data rate; Reference [18] proposed a novel optimally weighted non-coherent paradigm to combat the inter-symbol interference and then minimize the BER of a strong scattering plus time-varying channel response ultraviolet communication system; Reference [19] surveyed the co-deployment of a hybrid RF/optical or optical/optical wireless system in improving the individual system performance in terms of throughput, reliability, and energy efficiency.

In this paper, based on the former work on APD noise construction [13], and on the receiver signal modeling about the input-dependent and thermal noises in [15], we establish a new APD receiver signal model suitable for UWOC to thoroughly study the channel capacity of UWOC system with APD detector. Moreover, based on this new system model, the closed-form expressions of capacity lower bounds in APD receiving scenario are derived in detail, which mathematically disclose the relationship between the capacity bound and the pointing error angle. Furthermore, with maximizing the capacity bound as an optimization goal with respect to the titling receiver angle, we prove that this is in fact a simple convex optimization problem and can easily achieve the corresponding optimum solution. Theoretical analysis and simulation results both verify the effects of tilting the receiver plane on improving the system capacity.

The remainder of the paper is organized as follows: a system model suitable for describing the signal transmission in clear ocean water type UWOC with APD receiver and channel path loss together with pointing error is established in Section 2. In Section 3, based on the system model presented above, theoretical closed-form expressions of the capacity lower bounds for UWOC with APD are derived in detail under some necessary constraints. In Section 4, an optimization problem is formulated and solved to improve the channel capacity of UWOC with APD by appropriately tilting the receiver's plane. Section 5 demonstrates a series of numerical simulations to verify our mathematical optimization problem; and finally, in Section 6, we conclude the paper with some concise remarks.

## 2. System Model

A spatially Cartesian coordinate system for UWOC can be established as shown in the following Figure 1: point light source $O$ is assumed to be located at the origin of the coordinates; the APD receiver moves on a quarter circle plane with radius $r$ m (for simplicity of analysis, only quarter circle plane is considered and the quadrant where the receiver is located is set as the first one) in clear ocean water environment with related typical set of absorption and scattering coefficients of $(a, b) = (0.069, 0.08)$ m$^{-1}$ [3]; the horizontal distance between the light source and the circular surface is $D$ m, and the coordinates of the light source and receiver are [0,0,0] and $[x_0, y_0, z_0]$ respectively; the field of view (FOV)

of the receiver is 180°, and $d$ is the separation between the light source and the receiver. Please also note that, the distance parameters set in Figure 1 are determined by channel path simulations; one can find more details in Section 5.

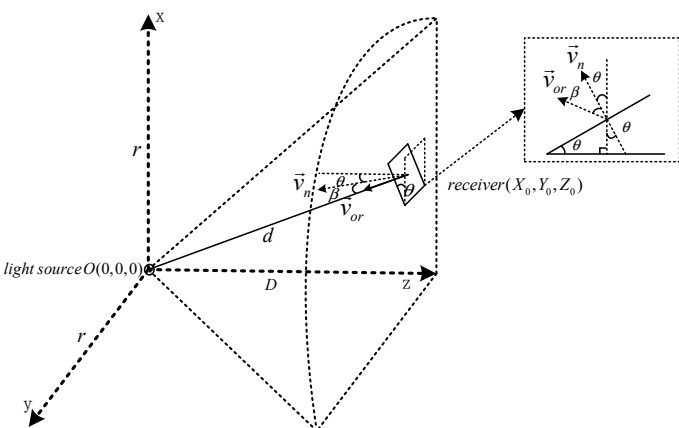

**Figure 1.** Geometric configuration model for UWOC system.

We additionally assume that the beam of the light source is always aimed at the receiver. For the receiver, the vector from the receiver to the light source $O$ is $\vec{V}_{or}$, $\vec{V}_n$ is the unit normal vector perpendicular to the receiver plane, the angle between the tilted receiver plane and the horizontal X-Y circular surface is $\theta$, and the angle between the normal vector $\vec{V}_n$ and pointing vector $\vec{V}_{or}$ is set to $\beta$. To further simplify the analysis, we have set the Z axis, $\vec{V}_n$ and $\vec{V}_{or}$ coplanar. In this way, the relationship between them can be listed in the dashed box in the upper-right corner of Figure 1. In addition, we ignore the influence of the size of the receiver on $\vec{V}_{or}$, that is, once the receiver's coordinate $[x_0, y_0, z_0]$ is fixed, the direction of $\vec{V}_{or}$ will not change with the tilting of the receiver plane. However, the direction of $\vec{V}_n$ will obviously change with the tilting of the receiver, that is, the pointing error angle $\beta$ between $\vec{V}_n$ and $\vec{V}_{or}$ will change, which should be a function of parameter $\theta$.

Assuming further that the transmitter uses the simple on-off keying (OOK) intensity modulation to transmit useful signals while the receiver adopts direct detection to make decisions, the received current signal of an APD detector after considering the channel path loss, the photo-current shot noise and thermal noise [13,15], has the following expression as

$$Y = GL\tau x + \sqrt{GL\tau x}\,n' + n, \tag{1}$$

where $G$ is the average gain of APD; $L$ is the channel path loss in clear ocean water; $\tau = \eta e/hv$ is the photoelectric transformation coefficient, in which $\eta$ is the quantum efficiency, $h$ is the Planck's constant, $v$ is the frequency of light wave in seawater, and $e$ is the electron charge; $x$ denotes the transmitted light-intensity signal, i.e., the OOK symbol, which is assumed to be drawn equal-probably from an OOK modulation constellation, that is, $x \in \{0, P_s\}$, where $P_s$ is the average transmitting power; $\sqrt{GL\tau x}\,n'$ is the so-called photo-current shot noise term [13], which is proportional to the received average current signal value, and $n'$ is a Gaussian white noise random variable with mean zero and variance $2eBGF_1$, in which $B$ is the filter bandwidth, $F_1 = \xi G + (2 - 1/G)(1 - \xi)$ is the excess noise factor, and $\xi$ is the ionization ratio; $n$ represents the background noise of APD, which is independent of noise $n'$ and can be modeled by the thermal noise of the load resistance of amplifier circuit as a Gaussian white noise term with mean zero and variance $\sigma^2$, i.e., reference [13]

$$\sigma^2 = \frac{4KTBF}{R}, \tag{2}$$

where $K$ is the Boltzmann constant, $T$ is the temperature in Kelvin, $B$ is the filter bandwidth mentioned above, $R$ is the resistance value of the load, and $F$ is the noise factor of the system. For simplicity of description below, we further record the ratio of the variance of input-dependent noise term $n'$ to that of the background noise term $n$ as $\varsigma^2$, where $\varsigma^2 = 2eBGF_1$, so the variance of input-dependent noise term $n'$ can be simply expressed as $\sigma^2\varsigma^2$. Please note that although the spectral efficiency of OOK modulation is only 1bit/s/Hz compared with other higher order modulation schemes [20], for example, quadrature phase shift keying (QPSK), $M$-ary quadrature and amplitude modulation ($M$-QAM), etc., however, its simple detection method and easy implementation for intensity modulation make it widely adopted and used in the research of optical communication community. In fact, due to the very high carrier frequency of light beam, the transmission data rate of UWOC system even with OOK modulation can reach Giga bits/second with quite low BER detection performance [3,6].

In (1), the channel path loss $L$ can be described as [4]

$$L = \tau_{channel} \cdot \tau_{point}, \tag{3}$$

where $\tau_{channel}$ is the channel loss, which is from the absorption and scattering due to seawater. Compared with the conventional Beer's law, a double-exponential channel loss model can more accurately depict the channel fading in a clear ocean water type [14,21]. Therefore, it can be modified and depicted as

$$\tau_{channel} = C_1 e^{C_2 d} + C_3 e^{C_4 d}, \tag{4}$$

where $C_1$, $C_2$, $C_3$, and $C_4$ are the fitting coefficients obtained by Monte Carlo simulations; $d$ is the propagation path from the light source to receiver depicted in Figure 1. However, when misalignment deployment of the receiver and transmitter occurs, the pointing loss must be considered. As shown in [5], this loss can be expressed as

$$\tau_{point} = cos\beta, \tag{5}$$

where $\beta$ is the pointing error angle in Figure 1. Inserting (4) and (5) into (3), the double-exponential channel loss model with pointing error can be written as [22]

$$L = \left(C_1 e^{C_2 d} + C_3 e^{C_4 d}\right) cos\beta. \tag{6}$$

According to the knowledge of spatial analytic geometry, the pointing error loss term $cos\beta$ can be expressed in the following as

$$cos\beta = \frac{\langle \vec{V}_n, \vec{V}_{or} \rangle}{\left\| \vec{V}_n \right\| \cdot \left\| \vec{V}_{or} \right\|}, \tag{7}$$

where vector $\vec{V}_n = [cos\varphi sin\theta, sin\varphi sin\theta, cos\theta]$, and $\vec{V}_{or} = [(0 - x_0), (0 - y_0), (0 - z_0)]$; substituting them into (7), it simplifies to

$$cos\beta = \frac{1}{d}[(-x_0)cos\varphi sin\theta + (-y_0)sin\varphi sin\theta + (-z_0)cos\theta], \tag{8}$$

where $\varphi$ is the azimuth angle formed by the positive direction of axis $X$ and the projection of $\vec{V}_n$ on the horizontal plane; $[x_0, y_0, z_0]$ are the coordinates of the receiver. In fact, since $\vec{V}_n$, $\vec{V}_{or}$, and $\vec{Z}$ are coplanar, $\varphi$ is totally determined by the coordinates of the receiver, as shown in Figure 2. As can be seen from the figure, the azimuth angle $\varphi$ can be calculated by

$$\varphi = tan^{-1}\frac{y_0}{x_0}. \tag{9}$$

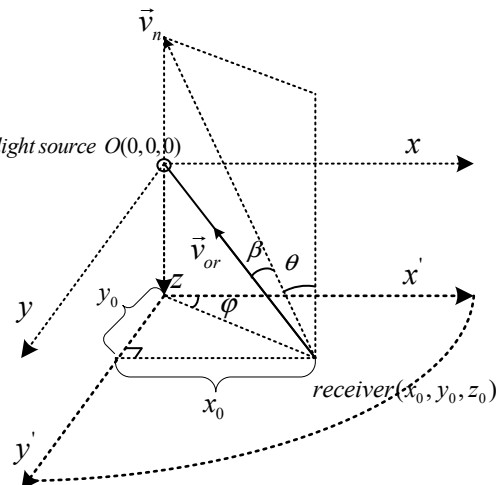

**Figure 2.** Illustration of projection of $\vec{V}_n$ on the first quadrant of X-Y plane.

In this UWOC system, due to the selected OOK modulation scheme, the light intensity signal $x$ defined in (1) should be restricted to a non-negative and real variable case, therefore, we have

$$x \geq 0. \tag{10}$$

Besides above, for the reasons of power consumption and practical implementation consideration, the average and peak powers of the transmitted OOK signal should further satisfy the following constrains

$$E[x] \leq \varepsilon, \tag{11}$$

and

$$P_r[x > P] = 0. \tag{12}$$

We refer to $\varepsilon$ as the allowed average power and $P$ the allowed peak one. Moreover, the ratio between the allowed average power and the allowed peak power is denoted by symbol $\alpha$, i.e.,

$$\alpha \triangleq \frac{\varepsilon}{P}, \tag{13}$$

which will be used below for derivation of the capacity bounds of UWOC system.

## 3. Derivation of the Lower Capacity Bounds for UWOC with APD

In order to simplify the math description, we set a scalar symbol $H$ as $H = GL\tau$. Thus, (1) is reduced to

$$Y = Hx + \sqrt{Hx}n' + n. \tag{14}$$

Then, the probability density function (PDF) of the received current signal $Y$, conditioned on the transmitted OOK symbol $x$, which is recovered by the APD receiver, can be given by

$$W(Y|x) = \frac{1}{\sqrt{2\pi\sigma^2(1+Hx\varsigma^2)}}e^{-\frac{(y-Hx)^2}{2\sigma^2(1+Hx\varsigma^2)}}. \tag{15}$$

According to the complicated constraints given in (10) to (12), and the biggest differential entropy conditions shown in [19], the classic Shannon channel capacity formula cannot be utilized in (14) anymore. That is, we need to re-derive the channel capacity of the UWOC system described by (14) under the constraints of (10) to (12). Generally speaking, it is a very hard issue to give an exact capacity formula to the channel mentioned above [15], so we circumambulate the issue and try to give an alternative solution, i.e., a tight lower

bound on the channel capacity, by using the knowledge of Shannon information theory. From the definition of channel capacity, we have the following inequality, i.e.,

$$C \geq I(Q, W)|_{any\ specified\ Q(\cdot)}$$

$$= h(Y) - h(Y|x), \tag{16}$$

where $C$ is the target channel capacity; $I(Q, W)$ stands for the mutual information between the input $x$ and the output $Y$ of the channel with PDF $W(\cdot|\cdot)$ shown in (15) when the input $x$ has distribution $Q(\cdot)$; $h(Y)$ is the output entropy and $h(Y|x)$ is the conditional entropy.

By the definition of differential entropy [17] and (15), $h(Y|x)$ can be obtained as

$$h(Y|x) = \frac{1}{2}E_Q\left[log 2\pi e\sigma^2\left(1 + Hx\varsigma^2\right)\right]$$

$$= \frac{1}{2}log(2\pi e\sigma^2\varsigma^2 H) + \frac{1}{2}E_Q[log x] + \frac{1}{2}E_Q[log(1 + \frac{1}{H\varsigma^2 x})], \tag{17}$$

where, $E_Q[\cdot]$ means calculating the mathematical expectation of the expression consisting of variable $x$ in brackets through the variable's probability distribution $Q(\cdot)$. (17) shows us that the entropy $h(Y|x)$ is a function of distribution $Q(\cdot)$.

In order to solve the lower bound on capacity in (16), we need to further manipulate $h(Y)$ into the function of $Q(\cdot)$. According to the work in [15], we can similarly get the lower bound of $h(Y)$ in terms of $Q(\cdot)$. To this goal, we first rewrite (14) as

$$Y = \underbrace{Hx + \sqrt{Hx}n'}_{Y_1} + n = Y_1 + n, \tag{18}$$

where $Y_1$ and $n$ are independent variables, satisfying $Y_1 \sim N(Hx, Hx\sigma^2\varsigma^2)$ and $n \sim N_R(0, \sigma^2)$. By the fact that conditioning will reduce entropy, we have

$$h(Y) = h(Y_1 + n) \geq h(Y_1 + n|n) = h(Y_1|n) = h(Y_1). \tag{19}$$

Hence, the problem of (16) is reduced to finding a lower bound to $h(Y_1)$.

According to the average light power constraint given in (11), we have an exponential distribution $Q_{Exp}(\cdot)$ on input $x$ to maximize the entropy $h(Y_1)$ [23], i.e.,

$$Q_{Exp}(x) \triangleq \frac{1}{\varepsilon}e^{-\frac{x}{\varepsilon}}, \qquad x \geq 0, \tag{20}$$

where $\epsilon$ is the input power constraint value. Since the conditional PDF of $Y_1$ on $x$ is given by

$$W_1(Y_1|x) = \frac{1}{\sqrt{2\pi Hx\sigma^2\varsigma^2}}e^{-\frac{(y_1 - Hx)^2}{2Hx\sigma^2\varsigma^2}}, \tag{21}$$

and if we choose $Q_{Exp}(\cdot)$ as the input distribution to our reduced channel $Y_1$, the corresponding output distribution is

$$(Q_{Exp}W_1)(y_1) = \frac{1}{\sqrt{\varepsilon H(\varepsilon H + 2\sigma^2\varsigma^2)}}exp\left(\frac{\sqrt{\varepsilon H}y_1 - \sqrt{\varepsilon H + 2\sigma^2\varsigma^2}|y_1|}{\sigma^2\varsigma^2\sqrt{\varepsilon H}}\right), \ y_1 \in R. \tag{22}$$

By using the data processing theorem for relative entropy [15], we obtain the following inequality

$$h(Y) = h(Y_1) \geq h(x) + f(\varepsilon, H), \tag{23}$$

where

$$f(\varepsilon, H) = \frac{1}{2}logH + \frac{1}{2}log\frac{\varepsilon H + 2\sigma^2\varsigma^2}{\varepsilon} - \frac{\varepsilon H + \sigma^2\varsigma^2}{\sigma^2\varsigma^2} + \frac{\sqrt{\varepsilon H(\varepsilon H + 2\sigma^2\varsigma^2)}}{\sigma^2\varsigma^2}, \qquad (24)$$

is a function which only relates to parameters $\varepsilon$ and $H$, and has noting to do with input $x$. The detailed derivations of (22) and (23) can be found in Appendix A.

Substituting (23), (24), and (17) into (16), we have

$$C \geq h(x) + f(\varepsilon, H) - \frac{1}{2}E_Q\left[log2\pi e\sigma^2\left(1 + Hx\varsigma^2\right)\right]$$

$$= h(x) + f(\varepsilon, H) - \frac{1}{2}log2\pi e\sigma^2\varsigma^2 H - \frac{1}{2}E_Q[logx] - \frac{1}{2}E_Q[log(1 + \frac{1}{H\varsigma^2 x})]. \qquad (25)$$

Till now, the lower bound of channel capacity about UWOC with APD detector can be calculated by choosing an input distribution $Q(\cdot)$ which maximizes the entropy $h(x)$ under the constraints (10), (11), and (12). This can be solved by applying the Lagrange multipliers approach. In addition, we consider the following two different cases of $\alpha$ value range:

1.  $\alpha \in \left(0, \frac{1}{3}\right)$. Both the average and peak power constraints are imposed in this case, then the input distribution $Q(\cdot)$ is expressed as

$$Q(x) \triangleq \begin{cases} \frac{\sqrt{\mu}}{\sqrt{P\pi x}erf(\sqrt{\mu})}e^{-\frac{\mu}{P}x}, & 0 < x \leq P \\ 0, & else \end{cases} \qquad (26)$$

where $\mu$ is the solution to equation

$$E[x] = \frac{P}{2\mu} - \frac{Pe^{-\mu}}{\sqrt{\mu}\sqrt{\pi}erf(\sqrt{\mu})} = \alpha P, \qquad (27)$$

in which, $erf(t) = \frac{2}{\sqrt{\pi}}\int_0^t e^{-u^2}du$ denotes the error function. The detailed derivations of (26) and (27) can be found in Appendix B. Please note that, since the function $f(\mu) = \frac{1}{2\mu} - \frac{e^{-\mu}}{\sqrt{\mu}\sqrt{\pi}erf(\sqrt{\mu})}$ is monotonically decreasing in $(0, \infty)$, i.e., it tends to 1/3 for $\mu \downarrow 0$ and to 0 for $\mu \uparrow \infty$ [15], the average to peak power ratio $\alpha$ satisfies $\alpha \in \left(0, \frac{1}{3}\right)$. With (26), we can solve the component terms with respect to the input signal $x$ in (25) as

$$h(x) = -\int_0^P Q(x)logQ(x)dx = \frac{1}{2}log\frac{P}{\mu} + log\sqrt{\pi}erf(\sqrt{\mu}) + \alpha\mu + \frac{1}{2}E_Q[logx], \quad (28)$$

and

$$\frac{1}{2}E_Q[log(1 + \frac{1}{H\varsigma^2 x})] \leq \frac{2\sqrt{\mu}}{\sqrt{\pi}erf(\sqrt{\mu})\sqrt{\varsigma^2 PH}}tan^{-1}\sqrt{\varsigma^2 PH} +$$

$$\frac{\sqrt{\mu}}{\sqrt{\pi}erf(\sqrt{\mu})}log\left(1 + \frac{1}{\varsigma^2 PH}\right). \qquad (29)$$

The detailed derivations of (29) can be found in Appendix C. Combining (25) with (24), (28), and (29), the lower capacity bound of UWOC with APD in case I can be obtained as

$$C \geq C_{Low} = \frac{1}{2}log\frac{P}{\mu} + log\sqrt{\pi}erf(\sqrt{\mu}) - \frac{1}{2}log2\pi e\sigma^2\varsigma^2 + \alpha\mu - 1 + f(H), \qquad (30)$$

where

$$f(H) = \frac{1}{2}log\frac{\varepsilon H + 2\sigma^2\varsigma^2}{\varepsilon} - \frac{\varepsilon H}{\sigma^2\varsigma^2} + \frac{\sqrt{\varepsilon H(\varepsilon H + 2\sigma^2\varsigma^2)}}{\sigma^2\varsigma^2} -$$

$$\frac{2\sqrt{\mu}}{\sqrt{\pi}erf\left(\sqrt{\mu}\right)\sqrt{\varsigma^2PH}}tan^{-1}\sqrt{\varsigma^2PH} - \frac{\sqrt{\mu}}{\sqrt{\pi}erf\left(\sqrt{\mu}\right)}log\left(1 + \frac{1}{\varsigma^2PH}\right). \tag{31}$$

2. $\alpha \in \left[\frac{1}{3}, 1\right]$. In this second case, the average and peak power constraints (11) and (12) are still satisfied. However, due to that $\alpha$ tends to $1/3$ for $\mu \downarrow 0$, the input distribution $Q(\cdot)$ for this second $\alpha$ case is reduced to the following expression by approaching $\mu$ in (26) to zero [15]

$$Q(x) \triangleq \begin{cases} \frac{1}{\sqrt{4Px}}, & 0 < x \le P \\ 0, & else \end{cases}. \tag{32}$$

Similarly, under this new distribution, we have

$$h(x) = logP - 1 + log2, \tag{33}$$

and

$$\frac{1}{2}E_Q[log\left(1 + H\varsigma^2x\right)] = \frac{1}{2}log\left(1 + H\varsigma^2P\right) - 1 + \frac{1}{\sqrt{H\varsigma^2P}}tan^{-1}\sqrt{H\varsigma^2P}. \tag{34}$$

By rewriting (25), that is,

$$C \ge h(x) + f(\varepsilon, H) - \frac{1}{2}E_Q\left[log2\pi e\sigma^2\left(1 + H\varsigma^2x\right)\right]$$

$$= h(x) + f(\varepsilon, H) - \frac{1}{2}log2\pi e\sigma^2 - \frac{1}{2}E_Q[log\left(1 + H\varsigma^2x\right)], \tag{35}$$

the lower capacity bound is

$$C_{Low} = log2P - 1 - \frac{1}{2}log2\pi e\sigma^2 + f(H), \tag{36}$$

where

$$f(H) = \frac{1}{2}log(H) + \frac{1}{2}log\frac{\varepsilon H + 2\sigma^2\varsigma^2}{\varepsilon} - \frac{\varepsilon H}{\sigma^2\varsigma^2} + \frac{\sqrt{\varepsilon H(\varepsilon H + 2\sigma^2\varsigma^2)}}{\sigma^2\varsigma^2}$$

$$- \frac{1}{2}log\left(1 + H\varsigma^2P\right) - \frac{1}{\sqrt{H\varsigma^2P}}tan^{-1}\sqrt{H\varsigma^2P}. \tag{37}$$

## 4. Optimization Problem Raising and Solving

Based on (6), (14), (30), and (36) mentioned above, it can be seen that once the distance $d$ is fixed, the capacity lower bound of UWOC is a unary function of the tilting angle $\theta$. Changing $\theta$ might obtain the maximum value of the lower bound of UWOC at some given distance, so the above question turns into one mathematical optimization problem. For this purpose, the optimization problem of capacity lower bound is first raised in this section, then the optimization issue is further proved to be a simple convex optimization one, and finally the theoretical solution to the optimal tilting angle is deduced.

### 4.1. Description of the Capacity Optimization Problem

Taking maximizing the capacity lower bound of the UWOC system shown in Figure 1 as an optimization target, and considering the limit of the tilting angle $\theta$ of the receiver, the optimization problem can be expressed as

$$\max_{\theta} C_{Low}, \ s.t. \ 0 \le \theta \le \pi/2 \tag{38}$$

### 4.2. Solution to the Optimization Problem

For the Case 1 that is, $\alpha \in \left(0, \frac{1}{3}\right)$, combining (30) with (31), the first derivative of the $C_{Low}$ with respect to variable $H$ is obtained as

$$\frac{dC_{Low}}{dH} = \frac{1}{2} \cdot \frac{\varepsilon}{\varepsilon H + 2\sigma^2 \varsigma^2} + \underbrace{\frac{\varepsilon^2 H + \varepsilon \sigma^2 \varsigma^2}{\sigma^2 \varsigma^2 \sqrt{\varepsilon H(\varepsilon H + 2\sigma^2 \varsigma^2)}} - \frac{\varepsilon}{\sigma^2 \varsigma^2}}_{①} +$$

$$\frac{\sqrt{\mu}}{\sqrt{\pi} erf(\sqrt{\mu})} \cdot \frac{1}{H\sqrt{\varsigma^2 PH}} \cdot tan^{-1}\sqrt{\varsigma^2 PH} +$$

$$\underbrace{\frac{\sqrt{\mu}}{\sqrt{\pi} erf(\sqrt{\mu})} \cdot \frac{1}{\varsigma^2 PH^2 + H} - \frac{\sqrt{\mu}}{\sqrt{\varsigma^2 PH}\sqrt{\pi} erf(\sqrt{\mu})} \cdot \frac{\varsigma^2 P}{\sqrt{\varsigma^2 PH}(1 + \varsigma^2 PH)}}_{② = 0}. \quad (39)$$

For expression ① above, we have

$$① = \frac{(\varepsilon H + \sigma^2 \varsigma^2) - \sqrt{\varepsilon H(\varepsilon H + 2\sigma^2 \varsigma^2)}}{\sqrt{\varepsilon H(\varepsilon H + 2\sigma^2 \varsigma^2)}} \cdot \frac{\varepsilon}{\sigma^2 \varsigma^2}. \quad (40)$$

In order to prove $(40) \geq 0$, we just need to simply prove

$$\varepsilon H + \sigma^2 \varsigma^2 \geq \sqrt{\varepsilon H(\varepsilon H + 2\sigma^2 \varsigma^2)}, \quad (41)$$

which is obviously identical to prove that

$$\varepsilon^2 H^2 + 2\varepsilon H\sigma^2 \varsigma^2 + \left(\sigma^2 \varsigma^2\right)^2 \geq \varepsilon^2 H^2 + 2\varepsilon H\sigma^2 \varsigma^2. \quad (42)$$

Obviously, the inequality (42) is satisfied. Since $H$ and $\varepsilon$ are nonnegative, we can easily come to the following conclusion

$$\frac{dC_{Low}}{dH} \geq 0, \ \alpha \in (0, 1/3). \quad (43)$$

As to the Case 2: $\alpha \in \left[\frac{1}{3}, 1\right]$, similarly, making a derivative of (36) with respect to $H$, and after some necessary manipulations, we have

$$\frac{dC_{Low}}{dH} = \frac{1}{2H} + \frac{1}{2} \cdot \frac{\varepsilon}{\varepsilon H + 2\sigma^2 \varsigma^2} + \underbrace{\frac{\varepsilon^2 H + \varepsilon \sigma^2 \varsigma^2}{\sigma^2 \varsigma^2 \sqrt{\varepsilon H(\varepsilon H + 2\sigma^2 \varsigma^2)}} - \frac{\varepsilon}{\sigma^2 \varsigma^2}}_{①} +$$

$$\frac{1}{2H\sqrt{\varsigma^2 PH}} \cdot tan^{-1}\sqrt{\varsigma^2 PH} - \frac{1}{2} \cdot \frac{\varsigma^2 P}{\varsigma^2 PH + 1} - \frac{1}{2\varsigma^2 PH} \cdot \frac{\varsigma^2 P}{\varsigma^2 PH + 1}$$

$$= \frac{1}{2} \cdot \frac{\varepsilon}{\varepsilon H + 2\sigma^2 \varsigma^2} + ① + \frac{1}{2H\sqrt{\varsigma^2 PH}} \cdot tan^{-1}\sqrt{\varsigma^2 PH} \geq 0, \ \alpha \in [1/3, 1]. \quad (44)$$

The last inequality is satisfied due to the fact that $① \geq 0$.

From both (43) and (44) discussed above, we can see that, no matter what the range of $\alpha$ is in, $C_{Low}$ is a monotonically increasing function of parameter $H$. Thus the optimization problem (38) can be written equivalently as

$$\max_{\theta} H, \ s.t. \ 0 \leq \theta \leq \pi/2. \quad (45)$$

Due to $H = GL\tau$, and $G$ and $\tau$ are positive constants, we further have

$$\max_{\theta} L, \quad s.t. \ 0 \leq \theta \leq \pi/2. \tag{46}$$

Substituting (8) into (6), the expression of $L$ with the tilting angle $\theta$ is

$$L = \left( C_1 e^{C_2 d} + C_3 e^{C_4 d} \right)[(-x_0)\cos\varphi\sin\theta +$$

$$(-y_0)\sin\varphi\sin\theta + (-z_0)\cos\theta]/d. \tag{47}$$

Further taking the second derivative of $L$ with respect to $\theta$ and after simplifying, it is easy to get $d^2 L/d\theta^2 = -L$. Since the channel fading $L$ is nonnegative, the second derivative of $L$ with respect to $\theta$ is less than or equal to 0, which indicates that the objective function $L$ is a convex function about $\theta$. In other words, there must exist a value of $\theta$ that maximizes $L$ and thus maximizes the capacity lower bound $C_{Low}$.

Since (47) is convex and the constraint is also convex, the optimization issue of (46) is a convex optimization problem. With the first derivative of $L$ with respect to $\theta$ being equal to zero, the optimal tilting angle $\theta_0$ for reaching the maximum lower capacity bound is

$$\theta_0 = \arctan[\frac{x_0\cos\varphi + y_0\sin\varphi}{z_0}] = \arctan[\frac{d'}{z_0}], \tag{48}$$

where $d'$ is the distance between the receiver and the projection point of light source on the *X-Y* circular plane (see Figure 2). Combining Figure 2 and (48), it is easy to see that the pointing error angle $\beta$ is equal to 0 degree while the lower capacity bound $C_{Low}$ obtains its maximum, that is, the optimal tilting angle $\theta_0$ is the case that the normal vector $\overrightarrow{V}_n$ and the incident light are fully aligned.

## 5. Numerical Simulations and Analyses

In this section, we will probe into how the distance $d$ between the receiver and light source, and the receiver's tilted angle $\theta$ influence the capacity lower bound. Meantime, the feasibility of overcoming the pointing error by tilting the receiver to improve the channel capacity of UWOC will also be investigated. Due to the limited space of the article, we will only discuss the Case 2 situation, i.e., the average-to-peak power ratio $\alpha$ being in interval $[1/3, 1]$. Moreover, according to the requirements of the simulation platform, without loss of generality, we set the horizontal distance $D$ between the light source and the circular surface in Figure 1 as 19.25 m, and the radius $r$ of the quarter circle is 12.6 m. That is, we achieve the channel loss simulation and the corresponding fitting coefficients $C_i$ in (4) based on the distance quantities set above. The key simulation parameters are summarized and listed in Table 1.

Figure 3 depicts the channel capacity lower bounds of UWOC versus the tilting angle $\theta$ when the distance $d$ between the light source and the receiver varies. As shown in Figure 3, although $d$ has different values, the variation trends of different curves with respect to the tilting angle $\theta$ are roughly consistent. It can be seen clearly from the figure that each curve has an optimal tilting angle, say $\theta_0$, relating to the maximum lower capacity bound at this distance. For example, if $d$ equals to 23 m, the maximum lower bound on channel capacity can be achieved when $\theta$ is set to 35 degrees, i.e., $\theta_0 = 35°$. Moreover, with the increase of the distance $d$, the corresponding optimal tilting angle gradually increases, which means that when the receiver is farther away from the light source, the receiver plane needs to be deflected at a larger angle to overcome the adverse effect of the pointing error. Therefore, we can reasonably tilt the receiver plane according to the optimal tilting angle under certain distance, and thus improve the channel capacity significantly. Furthermore, we also find that the inclination angle $\theta$ plays a dominant role in the capacity lower bound variation especially when $\theta$ is larger; while for smaller $\theta$ values, the distance $d$ is the main influence factor. We also noticed that the capacity curve of $d = 19.25$ m

seems considerably different from the others for tilting angles being larger than 75°. The reason for this distinction is due to that the receiver at this distance is just underneath the transmitting light source (see Figure 1), where the optimal tilting angle is zero. So, when the tilting angle exceeds 75°, the incident photons from the source are difficult to reach the receiver plane even the FOV = 180°; obviously, the received photons by the receiver when the tilting angle is 90°, that is, the receiver plane is vertical to the X-Y plane, will decrease to zero. This is why the capacity values of this curve decrease quickly after 75° and become zero when $\theta$ is equal to 90°.

**Table 1.** Main simulation parameters.

| Parameter | Symbol | Value | Parameter | Symbol | Value |
|---|---|---|---|---|---|
| Transmitting power | $P/P_s$ | 0.3 w | Planck's constant | h | $6.6 \times 10^{-34}$ J · s |
| Ionization ratio | $\xi$ | 0.06 | Speed of light in sea water | c | $2.26 \times 10^8$ m/s |
| Boltzmann constant | K | $1.38 \times 10^{-23}$ J/K | Filter bandwidth | B | 100 MHz |
| Load resistance | $R_L$ | 100 Ω | Temperature in Kelvin | T | 290 K |
| The noise factor of the system | F | 1 | Average gain of APD | G | 10 |
| Quantum efficiency | η | 0.82 | Electron charge | e | $1.6 \times 10^{-19}$ C |
| Fitting coefficient | $C_1$ | $0.9790 \times 10^{-4}$ | Fitting coefficient | $C_2$ | −0.1499 |
| Fitting coefficient | $C_3$ | $4.4162 \times 10^{-4}$ | Fitting coefficient | $C_4$ | −0.2876 |

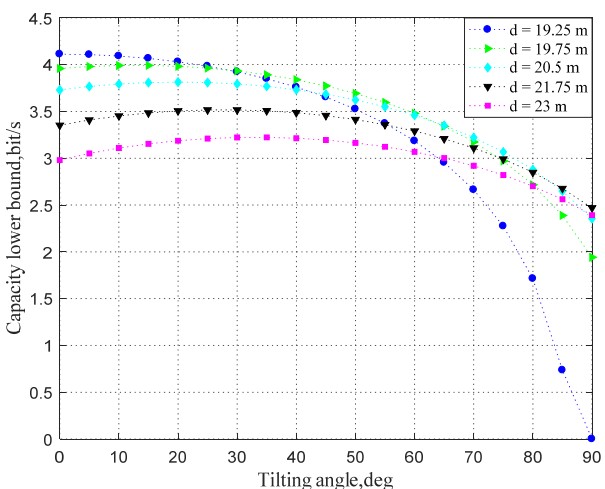

**Figure 3.** The capacity lower bounds versus tilting angle $\theta$ for UWOC.

Figure 4 shows the distribution of the system capacity lower bounds along with the receiver locations at the circular plane under before and optimally tilting the receiver. From Figure 4a,b, we can see that, whether tilting the receiver plane or not, the lower capacity bounds on the circumference of the boundary are the worst; and the maximum lower bounds on capacity can be obtained when the receiver is located directly below the light source. This result can be interpreted as that the channel fading $L$ gradually increases as distance $d$ increases, leading to capacity performance deterioration. However, after tilting the receiver plane according to the optimal tilting angle, the lower capacity bound of the entire X-Y plane (especially at the boundary) is greatly improved. For example, when $d = 23$ m, the lower bound is 2.9805 bit/s when the receiver plane is not inclined, while the capacity bound reaches 3.2240 bit/s after tilting with the optimal angle. This indicates that the tilted receiver can obviously eliminate the influence of the pointing error, and then optimize the channel performance of UWOC system.

Figure 5 shows the distribution of the optimal tilting angle when the receiver is located at different positions on the X-Y plane, where it is not difficult to find that the optimal tilting angles are the same on the circumference of any circle on the plane. Here is the possible reason: since the distances $d$ between the light source and the receiver on the same

circumference are the same, according to (46), (6), and (48), they have exactly the same optimization results for the tilting angle. As can be seen from Figure 5, when the receiver is set directly under the light source, we have the optimal tilting angle of 0°; while when the receiver is moving to the boundary, that is, the distance $d$ is going up, the optimal tilting angle will also increase and finally reach a maximum of 33.2°. These results just validate the conclusion of Figure 3.

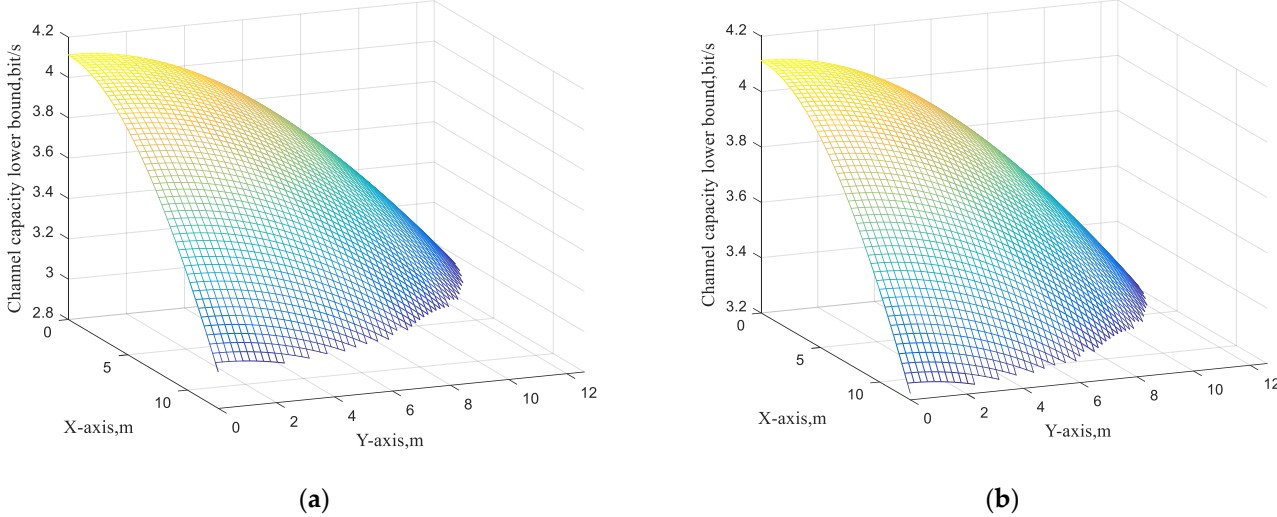

(**a**)                                       (**b**)

**Figure 4.** Capacity lower bounds with respect to the coordinates of the receiver (**a**) before tilting the receiver; (**b**) after optimally tilting the receiver.

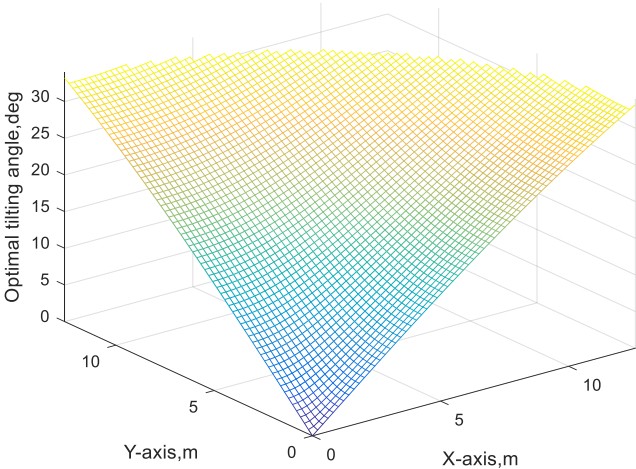

**Figure 5.** Optimal tilting angle distribution versus the receiver position on the X-Y plane.

## 6. Conclusions

In this paper, the effects of the tilted angle on the channel capacity of UWOC system with APD receiver adopting OOK modulation were analyzed. For this purpose, a mathematical system model suitable for describing the signal transmission in clear ocean water links with APD receiver and path loss with pointing error was established first; then closed-form expressions of the capacity lower bounds for UWOC with APD under the non-negativity, peak power, and average power constraints were derived in detail, which indicate that, no matter what the average-to-power ratios are, the capacity lower bounds are always monotonically increasing function with respect to the channel fading; based on these results, by relating the tilting angle of the receiver to the channel fading, an optimization problem of maximizing the channel capacity of UWOC system with APD via tilting

the receiver was proposed and solved theoretically. Both analysis and simulation results show that we can reasonably tilt the receiver's plane according to the solved optimum tilting angle, thus greatly improving the channel capacity of UWOC system.

**Author Contributions:** Conceptualization, M.J.; methodology, M.J. and H.S.; software, H.S.; validation, P.H. and G.T.; formal analysis, Y.L.; writing—original draft preparation, H.S.; writing—review and editing, Y.L. and M.J. All authors have read and agreed to the published version of the manuscript.

**Funding:** This research was funded by The Fundamental Research Funds for the Central Universities (No. 2019B15714) and the National Natural Science Foundation of China (No. 61832005).

**Conflicts of Interest:** The authors declare no conflict of interest.

## Appendix A. Proofs of Equations (22) and (23)

Combining (20) and (21), (22) can be rewritten as

$$(Q_{Exp}W_1)(y_1) = \int_0^\infty \left( \frac{1}{\varepsilon} e^{-\frac{x}{\varepsilon}} \right) \cdot \frac{1}{\sqrt{2\pi H x \sigma^2 \varsigma^2}} e^{-\frac{(y_1-Hx)^2}{2Hx\sigma^2\varsigma^2}} dx$$

$$\xrightarrow{Hx=x'} \int_0^\infty \frac{1}{\varepsilon H} e^{-\frac{x'}{\varepsilon H}} \cdot \frac{1}{\sqrt{2\pi x' \sigma^2 \varsigma^2}} e^{-\frac{(y_1-x')^2}{2x'\sigma^2\varsigma^2}} dx'$$

$$\xrightarrow{\varepsilon H=\varepsilon'} \int_0^\infty \frac{1}{\varepsilon'} e^{-\frac{x'}{\varepsilon'}} \cdot \frac{1}{\sqrt{2\pi x' \sigma^2 \varsigma^2}} e^{-\frac{(y_1-x')^2}{2x'\sigma^2\varsigma^2}} dx'$$

$$= \frac{1}{\sqrt{2\pi(\varepsilon')^2\sigma^2\varsigma^2}} \int_0^\infty (x')^{-\frac{1}{2}} \cdot e^{-\frac{2\sigma^2\varsigma^2(x')^2+\varepsilon'y_1^2+\varepsilon'(x')^2-2\varepsilon'y_1x'}{2\varepsilon'\sigma^2\varsigma^2x'}} dx'$$

$$= \frac{1}{\sqrt{2\pi(\varepsilon')^2\sigma^2\varsigma^2}} \cdot e^{\frac{y_1}{\sigma^2\varsigma^2}} \int_0^\infty (x')^{-\frac{1}{2}} \cdot e^{-\left(\frac{2\sigma^2\varsigma^2+\varepsilon'}{2\varepsilon'\sigma^2\varsigma^2}\right)\cdot x'} \cdot e^{-\frac{y_1^2}{2\sigma^2\varsigma^2}\cdot\frac{1}{x'}} dx'. \tag{A1}$$

According to the elementary function integral formula (3.471.15) in [24], that is,

$$\int_0^\infty (x)^{-\frac{1}{2}} \cdot e^{-ax} \cdot e^{-\frac{b}{x}} dx = \frac{\sqrt{\pi}e^{-2\sqrt{ab}}}{\sqrt{a}} \quad (a>0, b>0), \tag{A2}$$

and after some necessary manipulation and simplification, we achieve the final expression of (22).

The proof of (23) is based on the data processing inequality for relative entropies [23], i.e.,

$$D(Q||Q_{Exp}) \geq D(QW_1||Q_{Exp}W_1), \tag{A3}$$

where $(QW_1)(\cdot)$ denotes the $Y_1$ channel output distribution with an input distribution $Q(\cdot)$. By the definition of the relative entropy [23], the left-hand side of (A3) could be evaluated by

$$D(Q||Q_{Exp}) \triangleq \int_{-\infty}^\infty Q(x) log\left(\frac{Q(x)}{Q_{Exp}(x)}\right) dx. \tag{A4}$$

By (20), (A4) can be denoted as

$$D(Q||Q_{Exp}) = -h_Q(x) - E_Q[log(\frac{1}{\varepsilon}e^{-\frac{x}{\varepsilon}})]$$

$$= -h(x) + log(\varepsilon) + 1. \tag{A5}$$

With (22), the right-hand side of (A3) could be rewritten as

$$D(QW_1 \| Q_{Exp}W_1) \triangleq \int_{-\infty}^{\infty} QW_1(y) log(\frac{QW_1(y)}{Q_{Exp}W_1(y)}) dx$$

$$= -h(Y_1) + \frac{1}{2} log(\varepsilon H) + \frac{1}{2} log(\varepsilon H + 2\sigma^2\varsigma^2) - \frac{\varepsilon H}{\sigma^2\varsigma^2} +$$

$$\frac{1}{\sigma^2\varsigma^2} \sqrt{\frac{\varepsilon H + 2\sigma^2\varsigma^2}{\varepsilon H}} E_{(QW_1)}[|Y_1|]$$

$$\geq -h(Y_1) + \frac{1}{2} log(\varepsilon H) + \frac{1}{2} log(\varepsilon H + 2\sigma^2\varsigma^2) - \frac{\varepsilon H}{\sigma^2\varsigma^2} + \frac{\sqrt{\varepsilon H(\varepsilon H + 2\sigma^2\varsigma^2)}}{\sigma^2\varsigma^2}. \tag{A6}$$

Here we have used the Jensen inequality with the convex function $|\cdot|$ to get

$$E_{(QW_1)}[|Y_1|] \geq \left| E_{(QW_1)}[Y_1] \right| = \varepsilon H. \tag{A7}$$

Combining (A3), (A5) with (A6) yields (23) and (24).

**Appendix B. Proofs of Equations (26) and (27)**

The input distribution $Q(\cdot)$ maximizing the differential entropy $h(x)$ under the constraints of (10), (11), and (12), is with the following form [19] (Chapter 12)

$$Q(x) = e^{\lambda_0 + \lambda_1 x + \lambda_2 log x}, \tag{A8}$$

where $\lambda_0$, $\lambda_1$, and $\lambda_2$ are Lagrange factors needing to be optimized. Based on (10) and (12), and using the basic property of PDF function, we have

$$\int_0^P Q(x) dx = 1. \tag{A9}$$

Substituting (A8) into (A9), there is

$$e^{\lambda_0} \int_0^P e^{\lambda_1 x} \cdot x^{\lambda_2} dx = 1. \tag{A10}$$

By (11) and (A8), and using the integration by parts, we further get

$$\int_0^P x \cdot Q(x) dx = e^{\lambda_0} \left( \frac{P^{\lambda_2+1}}{\lambda_1} \cdot e^{\lambda_1 P} - \frac{\lambda_2+1}{\lambda_1} \int_0^P e^{\lambda_1 x} \cdot x^{\lambda_2} dx \right) = \varepsilon. \tag{A11}$$

Combining with (A10), (A11) results in

$$\frac{P^{\lambda_2+1}}{\lambda_1} \cdot e^{\lambda_0+\lambda_1 P} - \frac{\lambda_2+1}{\lambda_1} = \varepsilon. \tag{A12}$$

Specially, by assuming $\lambda_1 = -\frac{\mu}{P}$ and $\lambda_2 = -\frac{1}{2}$, (A10) could be denoted as

$$e^{\lambda_0} \int_0^P e^{-\frac{\mu}{P}x} \cdot x^{-\frac{1}{2}} dx = 1. \tag{A13}$$

Using the change of variable technique, that is, letting $\sqrt{x} = t$, (A13) changes into

$$2e^{\lambda_0} \int_0^{\sqrt{P}} e^{-\frac{\mu}{P}t^2} dt = 1. \tag{A14}$$

Utilizing the indefinite integral formula (2.33.16) in [24], we have

$$e^{\lambda_0} = \sqrt{\frac{\mu}{P\pi}} \cdot \frac{1}{erf\left(\sqrt{\mu}\right)}. \tag{A15}$$

Obviously, the fixed Lagrange factors $\lambda_1 = -\frac{\mu}{P}$, $\lambda_2 = -\frac{1}{2}$, and $e^{\lambda_0} = \sqrt{\frac{\mu}{P\pi}} \cdot \frac{1}{erf\left(\sqrt{\mu}\right)}$ should satisfy (A12). Substituting these three parameters into (A8) and (A12), and after some necessary manipulations, we have (26) and the following equation

$$\frac{P}{2\mu} - \frac{Pe^{-\mu}}{\sqrt{\mu}\sqrt{\pi}erf\left(\sqrt{\mu}\right)} = \varepsilon, \tag{A16}$$

i.e., $\mu$ is the solution to (A16).

**Appendix C. Proofs of Equation (29)**

According to (26), we have

$$\frac{1}{2}E_Q[log(1 + \frac{1}{H\varsigma^2 x})] = \frac{1}{2}\int_0^P log\left(1 + \frac{1}{H\varsigma^2 x}\right) \cdot \frac{\sqrt{\mu}}{\sqrt{P\pi x}erf\left(\sqrt{\mu}\right)} \cdot e^{-\frac{\mu}{P}x}dx. \tag{A17}$$

Because of

$$e^{-\frac{\mu}{P}x} \leq 1, \quad (P > 0, \ \mu > 0 \ and \ 0 < x \leq P), \tag{A18}$$

we can get the following derivation

$$\frac{1}{2}E_Q[log(1 + \frac{1}{H\varsigma^2 X})] \leq \frac{1}{2}\int_0^P log\left(1 + \frac{1}{H\varsigma^2 x}\right) \cdot \frac{\sqrt{\mu}}{\sqrt{P\pi x}erf\left(\sqrt{\mu}\right)}dx$$

$$\xrightarrow{x=x'} \frac{\sqrt{H}}{2H}\int_0^{HP} log\left(1 + \frac{1}{\varsigma^2 x'}\right) \cdot \frac{\sqrt{\mu}}{\sqrt{P\pi x'}erf\left(\sqrt{\mu}\right)}dx'$$

$$\xrightarrow{\sqrt{x'}=t} \frac{\sqrt{H}\sqrt{\mu}}{H\sqrt{P\pi}erf\left(\sqrt{\mu}\right)}\int_0^{\sqrt{HP}} log\left(1 + \frac{1}{\varsigma^2 t^2}\right)dt$$

$$= \frac{\sqrt{H}\sqrt{\mu}}{H\sqrt{P\pi}erf\left(\sqrt{\mu}\right)}\int_0^{\sqrt{HP}} [\underbrace{log\left(t^2 + \frac{1}{\varsigma^2}\right)}_{①} - \underbrace{log\ t^2}_{②}]dt. \tag{A19}$$

By using the indefinite integral formula (2.733.1) in [24], that is,

$$\int log\left(x^2 + a^2\right)dx = xlog\left(x^2 + a^2\right) - 2x + 2a \cdot tan^{-1}\frac{x}{a}, \tag{A20}$$

The integral results of expressions ① and ② are

$$① = \sqrt{PH}log\left(PH + \frac{1}{\varsigma^2}\right) - 2\sqrt{PH} + \frac{2}{\varsigma}tan^{-1}(\sqrt{PH\varsigma^2}), \tag{A21}$$

and

$$② = \sqrt{PH}log(PH) - 2\sqrt{PH}. \tag{A22}$$

Combining (A19), (A21), and (A22), and after simplification, we get (29).

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
