# Peer review of "Effect of the APD Receiver’s Tilted Angle on Channel Capacity for Underwater Wireless Optical Communications"

_electronics, doi:10.3390/electronics10182246_

Round 1

Reviewer 1 Report

The paper examines the effect of the APD-type receiver’s tilted angle on the lower bound of the capacity of an underwater wireless optical communication link. The paper proceeds by presenting a method for optimizing the receiver’s tilting angle with respect to the link’s capacity (the lower bound).

The paper provides sufficient background information and an adequate literature review and, though calculations are sometimes complicated, it presents its case in a clear and scientifically sound manner. Merits of the paper are the detailed calculations and closed-form results and the informative figures regarding simulation results.

I am only puzzled about the form of the “d=19,25m” curve which seems considerably different from the others for tilting angles larger than 75o. Perhaps, the authors could make a short comment on that.

The paper is well written from the language point of view – there are only a few spelling mistakes (e.g. the words “detailedly” in line 380 and “analyses” in line 384).

The article is publishable roughly in its present form (subject to minor spell checking and maybe addition of the comment mentioned above).

Reviewer 2 Report

The authors investigated the impact of the tilted angle in the receiver on the channel capacity of an underwater wireless optical communication system, where an avalanche photodiode detector is employed in the receiver. The results and discussions are interesting and helpful for optical wireless communication systems. This paper can be published in Electronics, provided following issues can be addressed.

  1. Please use the template provided by the journal
  2. Some abbreviations should be clarified when they appear for the first time.
  3. Specify more parameters used in the transmission setup.
  4. Add some discussion regarding the spectral efficiency and the modulation format of the considered optical communication systems.
  5. Supplement discussions on the state-of-the-art techniques in improving the performance of the optical wireless communication systems

See e.g.

Chowdhury et al., Optical wireless hybrid networks: trends, opportunities, challenges, and research directions, IEEE Communications Surveys & Tutorials, 2020.

Chi at al., Visible light communication in 6G: advances, challenges, and prospects, IEEE Vehicular Technology Magazine, 2020.

Hu et al., Non-coherent detection for ultraviolet communications with inter-symbol interference, Journal of Lightwave Technology, 2020.

Chaaban et al., Capacity of optical wireless communication channels, Philosophical Transactions A of the Royal Society, 2020.
